# A New Ultrasound Speckle Reduction Algorithm Based on Superpixel Segmentation and Detail Compensation

**Yang Chen, Ming Zhang, Hong-Mei Yan, Yong-Jie Li and Kai-Fu Yang ***

MOE Key Lab for Neuroinformation, University of Electronic Science and Technology of China,
Chengdu 610054, China; sophie-0627@163.com (Y.C.); zm_uestc@163.com (M.Z.); hmyan@uestc.edu.cn (H.-M.Y.);
liyj@uestc.edu.cn (Y.-J.L.)
* Correspondence: yangkf@uestc.edu.cn

**Abstract:** Speckle is a kind of noise commonly found in ultrasound images (UIs). Although traditional local operation-based methods, such as bilateral filtering, perform well in de-noising normal natural images with suitable parameters, these methods may break local correlations and, hence, their performance will be highly degraded when applied to UIs with high levels of speckle noise. In this work, we propose a new method, based on superpixel segmentation and detail compensation, to reduce UI speckle noise. In particular, considering that superpixel segmentation has the advantage of adhering accurately to the boundaries of objects or local structures, we propose a superpixel version of bilateral filtering to better protect the local structure during de-noising. Additionally, a human visual system (HVS)-inspired strategy for spatial compensation is introduced, in order to recover sophisticated edges as much as possible while weakening the high-frequency noise. Experiments on synthetic images and real UIs of different organs show that, compared to other methods, the proposed strategy can reduce ultrasound speckle noise more effectively.

**Keywords:** ultrasound image; denoising; superpixel; bilateral filter

## 1. Introduction

Ultrasound images (UI) are becoming more and more popular in clinical diagnosis due to their economical and practical advantages. However, due to the eliminative or constructive interference between different ultrasonic waves during the image generation phase, the collected UIs inevitably contain speckle-patterned noise. Affected by the speckle noise, UIs always exhibit low resolution, which raises the difficulty of identification and diagnosis in clinical practice. Furthermore, the accuracy of many tasks designed to assist diagnosis, such as image segmentation [1] or detection and classification [2], will also be degraded by the speckle noise. How to reduce the noise effects while effectively preserving the image details is becoming a challenging problem.

Many methods have been proposed to solve this problem, and most de-noising methods try to infer the true images based on the information redundancy character of the noisy images. For example, transform domain methods are mainly based on the assumption that an image can be sparsely represented by basic components, and so the noise is uniformly spread throughout the coefficients in the transform domain [3]. The useful details are concentrated in the largest parts of the high frequency components, while the noise exists in smaller ones; so, we can use some shrinking strategies to remove or reduce the noise. However, these kinds of methods face the risk of introducing pseudo-Gibbs artifacts when using a hard-shrinking strategy, or of over-smoothing when using a soft-shrinking strategy. As an interesting solution, Farouj et al. proposed a data-driven method to adapt the wavelet

threshold paradigm [4]. An extensive review of despeckling methods using various transformation techniques can be found in [5].

Along another line, the commonly used spatial domain methods [6–9] use local statistics to represent the information of the recovered images. These methods are often based on the multiplicative speckle model. The most successful methods among them are anisotropic diffusion [9] and the bilateral filter [6], which are all well-known due to their good ability for preserving edges. However, these kinds of methods corrupt the correlations between neighboring pixels, and thus perform badly for images with strong noise [3,10]. Inspired by the idea of decreasing pixel variation in homogenous regions while maintaining (or improving) the differences in the mean values of different regions, Tay et al. developed a squeeze box filter (SBF) to remove speckle noise [11].

Moreover, another group of sparse representation methods—namely non-local methods—has been proposed, in order to break the local dependence. In general, these methods use the property of information redundancy among similar patches to reduce the noise [12,13]. The common difficulty in these kinds of methods is how to find the candidate patches, as unsuitable candidate patches greatly reduce the de-noising performance, especially for the ultrasound images with strong noise. Recently, Zhu et al. [14] used a guidance image, derived from the windowed inherent variation measure [15], to overcome this problem. Furthermore, Yang et al. combined local and non-local properties and used the local statistics to help the selection of similar patches [16]. Santos et al. developed a modified version of the efficient block-matching collaborative filtering (BM3D) de-speckling method, based on the well-known statistical divergences [17]. Recently, Yaseen Jabarulla et al. proposed a multiplicative speckle suppression technique, based on sparse representation over dictionary learning [18].

Non-local methods always use a square window to search for similar patches. However, this strategy is not always suitable for edge regions. Additionally, the patch size needs to be pre-learned and then fixed for the test images. Based on the superpixel properties around the region edges, Li et al. [19] proposed a method of superpixel-guided non-local means for natural image noise reduction. Superpixel models have also been introduced to estimate the image noise level [20,21]. As the superpixels can be used to segment the images effectively, using the local statistics of superpixels to recover the images may be another feasible way for speckle noise reduction from ultrasound images.

In this paper, we propose a new ultrasound speckle noise reduction method, based on the strategies of superpixel reconstruction and spatial compensation. Specifically, we introduce the superpixel version of bilateral filtering to realize speckle noise reduction. Furthermore, a human visual system (HVS)-inspired model is proposed to realize the spatial compensation, in order to recover the sophisticated details effectively.

The rest of the paper is organized as follows. In Section 2, we introduce our proposed despeckling model. Then, we validate the effectiveness of our approach in Section 3 and, finally, present the discussion and conclusion in Section 4.

## 2. Proposed Superpixel-Based Model

The proposed method mainly consists of three steps: sparsification, reconstruction, and compensation. Figure 1 shows the flowchart of the proposed method.

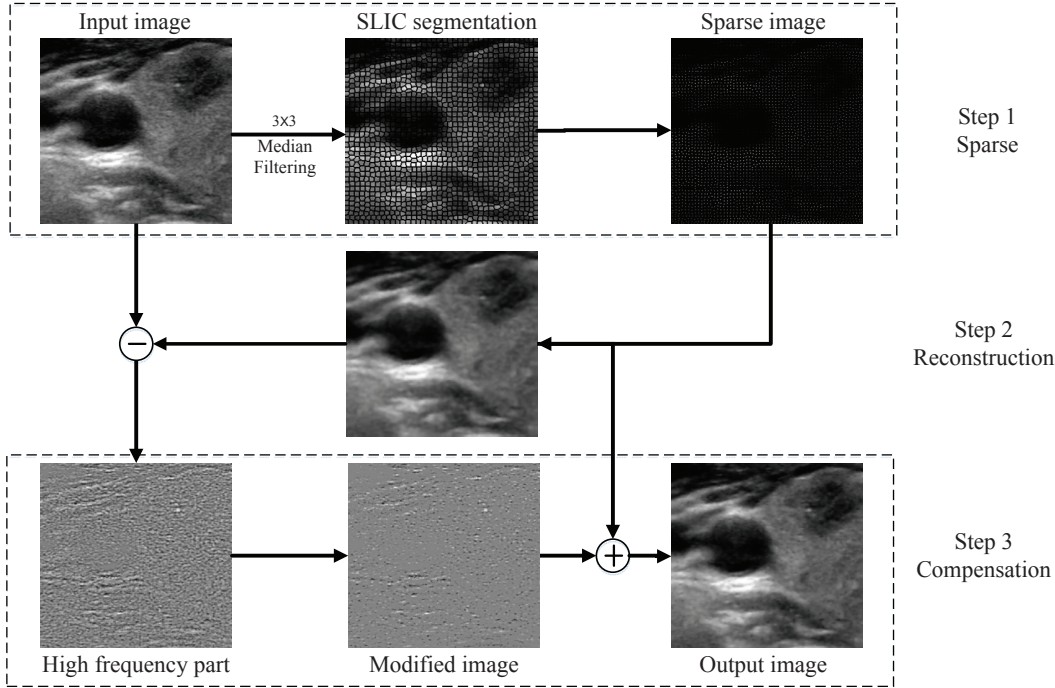

**Figure 1.** Flowchart of the proposed method.

### 2.1. Image Sparsification

As mentioned in Section 1, commonly used superpixel methods, such as the simple linear iterative clustering (SLIC)-based superpixel generating algorithm [22], have the advantage of adhering accurately to the boundaries of objects or local structures for images with low-level noise. However, the performance of SLIC segmentation will be highly degraded for ultrasound images which contain high-level noise. In order to always obtain robust SLIC segmentation for ultrasound images, we first adopt the median filter to pre-process the input ultrasound images. The purpose of this pre-processing is to reduce the influence of high-frequency noises (not speckles) and improve the robustness of SLIC segmentation. Therefore, we set the median filter to a small size; namely $3 \times 3$.

After the SLIC superpixel segmentation, we keep only the central points of all the superpixels and the label of each pixel and, for each superpixel, we reassign the value of its center as the median value of all the pixels within the superpixel. Now, we obtain a de-noised sparse distribution to sparsely represent the input image. We will use this sparse distribution, later, to reconstruct the low-frequency component of the given ultrasound image.

### 2.2. Image Reconstruction

Similar to the bilateral filtering, for each pixel at position $(i, j)$ that needs to be reconstructed, we use the following two measures as the weights to reconstruct its value: (1) the distance of the current pixel at $(i, j)$ to the center of its neighboring superpixel at $(k, l)$, and (2) the intensity difference between the center of the superpixel at $(i, j)$ and the center of its neighboring superpixel at $(k, l)$. Figure 2 shows a graphical illustration of the reconstruction, and the reconstructed image $I_{low}(i, j)$ can be mathematically obtained as

$$\omega(i, j, k, l) = \exp\left(-\frac{(i-k)^2 + (j-l)^2}{2\sigma_d^2} - \frac{||I(i,j) - I(k,l)||^2}{2\sigma_r^2}\right), \tag{1}$$

$$I_{low}(i, j) = \frac{\sum_{k,l} I(k,l)\omega(i,j,k,l)}{\sum_{k,l} \omega(i,j,k,l)}, \tag{2}$$

where $I(i, j)$ and $I(k, l)$ are, respectively, the intensity values for the centers of the superpixels at $(i, j)$ and $(k, l)$. It is clear that $\omega(i, j, k, l)$ represents the combined weight of the neighboring superpixel at $(k, l)$ with respect to the current pixel at $(i, j)$. As is typically done, we only pick the four neighboring superpixels in the up, down, left, and right directions, respectively. Furthermore, $\sigma_d$ and $\sigma_r$ are the scale parameters controlling the smoothing effect; higher values of which produce smoother reconstructed images. We empirically set $\sigma_d = 4$ and $\sigma_r = 0.3$ for all of the experiments in this work, as explained next.

In this work, we set appropriate values of $\sigma_d$ and $\sigma_r$ in Equation (1) to prevent the unbalance of the distance- and intensity-related weightings for image reconstruction instead of a simple normalization. This is mainly due to a technical consideration that the setting of $\sigma_d$ and $\sigma_r$ should depend on the superpixel size in this study. For example, according to the analysis and discussion in Section 3.1, we set the superpixel size as about 7 pixels in this work. Therefore, for the image reconstruction with four neighboring superpixels, the maximal distance between the current point and the neighboring superpixel centers (i.e., the term $\sqrt{(i-k)^2 + (j-l)^2}$ in Equation (1)) is about 10 pixels (along the diagonal line). Therefore, we set $\sigma_d = 4$ to control the decrease rate of the distance-related weighting. On the other hand, the maximum intensity difference between two pixels (i.e., the term $||I(i, j) - I(k, l)||$ in Equation (1)) is 1.0. Therefore, we set $\sigma_r = 0.3$ to control the decrease rate of the intensity-related weighting.

In summary, the setting of $\sigma_d = 4$ and $\sigma_r = 0.3$ used in this work can well balance the distance- and intensity-related weightings (i.e., with similar decreasing rates) when reconstructing the image with the superpixel size as about 7 pixels. Please note that as mentioned above, the setting of $\sigma_d$ and $\sigma_r$ should be dependent on the superpixel size. Large $\sigma_d$ and $\sigma_r$ values are required for the image segmentation with large superpixel sizes, and vice versa.

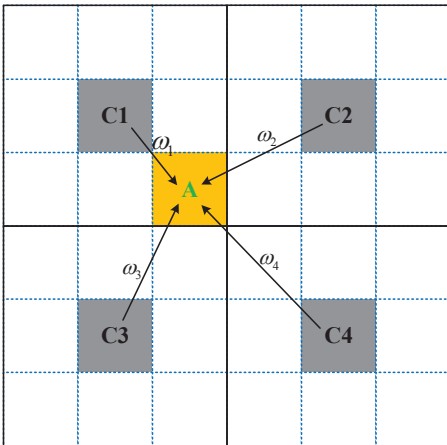

**Figure 2.** Illustration of the image reconstruction. A represents the current pixel to be reconstructed and C1, C2, C3, and C4 represent the four superpixel centers in a $3 \times 3$ squared grid around A.

## 2.3. Image Compensation

The operations described above can be regarded to be a superpixel version of bilateral filtering. It is clear that the image details with a scale smaller than the superpixel size will be over-smoothed incorrectly. To precisely maintain useful image details, we introduce a compensation operation. It should be noted that compensation is a commonly used strategy for fine-tuning ultrasound image details [14]. However, differing from others, our compensation strategy is HVS-inspired, which is described below.

As the reconstructed image can be regarded as the low frequency component of the input ultrasound image, we can subtract the low frequency component from the input image to get the high frequency component of the input image:

$$I_{high}(x,y) = I(x,y) - I_{low}(x,y). \tag{3}$$

This high frequency component, $I_{high}(x,y)$, contains both the useful details and the unwanted high-frequency noise, which are difficult to separate from each other. In this work, we propose a HVS-inspired solution to separate and remove the unwanted high-frequency noise from the useful details of $I_{high}(x,y)$. Our solution is based on the following evidence from the field of visual neuroscience: (1) As the first stage of the HVS, the retina processes the visual information, from the bipolar cells to the ganglion cells, along the ON and OFF pathways. (2) The ON and OFF pathways process the bright and dark components, respectively. (3) The responses of the bipolar and ganglion cells in the retina show center-surround opponent patterns within their receptive fields (RFs).

Inspired by these biological findings, we first separate the high-frequency component into bright and dark parts, which are written as

$$I_b(x,y) = I_{high}(x,y), \quad where \ I_{high}(x,y) >= 0 \tag{4}$$

$$I_d(x,y) = -I_{high}(x,y), \quad where \ I_{high}(x,y) < 0. \tag{5}$$

Then, we use the non-classical receptive field (nCRF) model to separately process these two parts of the high frequency component. Suggested by the physiological finding that some retinal ganglion cells (RGCs) in cats respond quite weakly to the stimulus of dispersedly distributed dots, in comparison to compactly distributed dots (e.g., where the dispersedly distributed dots are compacted into a line) [23], we have built a nCRF model based on RGCs, in our previous works, for the tasks of image de-hazing and color constancy [24,25]. As for the RF's spatial structure, a nCRF model of ON-type contains an excitatory center and an inhibitory surround with a disinhibition property (see Figure 3)—that is, the RF surround is composed of many sub-units, and these sub-units first inhibit each other; then, the inhibited sub-units in the RF surround inhibit the neuronal response elicited by the stimulus in the RF center. The neuronal processing can be formulated as follows.

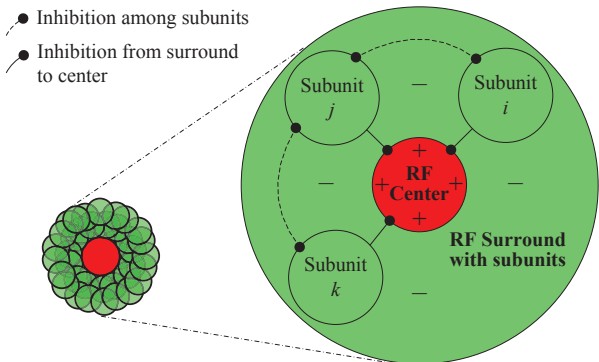

**Figure 3.** Illustration of the spatial structure of the non-classical receptive field (nCRF) model of ON-type, containing an excitatory center and an inhibitory surround with multiple sub-units.

We, first, define a two-dimensional (2D) Gaussian function with a scale of $\sigma$ as

$$g(x,y;\sigma) = \frac{1}{2\pi\sigma^2} \exp(-(x^2 + y^2)/(2\sigma^2)). \tag{6}$$

For the bright part of the high-frequency component (i.e., $I_b(x,y)$), the ON-type nCRF model is employed to compute the neuronal response as

$$S_{on}(x,y) = \mathbb{H}\{I_b(x,y) \otimes (g(x,y;\sigma_c) - g(x,y;\sigma_s))\}, \tag{7}$$

$$SS_{on}(x,y) = \sum_{(i,j) \in surround} S_{on}(i,j), \tag{8}$$

$$R_{on}(x,y) = \mathbb{H}\{I_b(x,y) - w_{on}(x,y) \cdot SS_{on}(x,y)\}, \tag{9}$$

where $S_{on}(x,y)$ is strength of the subunit at $(x,y)$ after being inhibited by its neighboring subunits, using the convolution operation with a difference of Gaussian (DoG) kernel; $\sigma_c$ and $\sigma_s$ are, respectively, the scales of the two Gaussians (in this study, we experimentally set $\sigma_c = 4$ and $\sigma_s$ as three times $\sigma_c$, based on electrophysiological observations [26]); $\otimes$ denotes the convolution operation; and $\mathbb{H}\{\}$ is used to guarantee that the neuronal responses should not be negative (i.e, $\mathbb{H}\{s\} = s$ while $s > 0$ and $\mathbb{H}\{s\} = 0$ while $s \leq 0$). According to the nCRF model, obtaining negative signals after surround inhibition means that these pixels receive strong inhibition and, hence, they should be regarded as speckles and be removed. Therefore, setting the negative responses to zeros does not just achieve despeckling, but also avoid the further processing of those potential interference of negative signals.

It is clear that $SS_{on}(x,y)$ is the total strength of the whole RF surround, and that $R_{on}(x,y)$ is the final neuronal response by subtracting the surround modulation from the response of RF center. Furthermore, $w_{on}(x,y)$ is a spatially varying weight, used to control the contribution of surround inhibition to the center, which is computed as

$$w_{on}(x,y) = m/CC_{on}(x,y), \tag{10}$$

where $m$ is a (global) constant and $CC_{on}(x,y)$ is the mean of $I_b(x,y)$ within the RF (including the center and its surround) centered at $(x,y)$. For all the synthetic images tested in this work, we set $m = 0.2$, and, for all the real ultrasound images, we set $m = 0.05$. Let us, first, assume that the noise is at the same level across the whole image. Hence, there would be more useful signals within the brighter regions with higher $I_b(x,y)$ values, and a higher $I_b(x,y)$ results in a higher $CC_{on}(x,y)$ and, hence, a lower $w_{on}(x,y)$, which contributes to the preservation of the useful signals by reducing the surround inhibition. Meanwhile, there would be less useful signals within the darker regions with lower $I_b(x,y)$ values, which results in a higher $w_{on}(x,y)$ and, hence, stronger inhibition of the unwanted signals within these regions. Such spatially adaptive inhibition makes the nCRF-based compensation work well in preserving details and suppressing noise.

Similarly, for the dark part of the high-frequency component (i.e.,$I_d(x,y)$), the OFF-type nCRF model is employed to compute the neuronal response $R_{off}(x,y)$, as

$$S_{off}(x,y) = \mathbb{H}\{I_d(x,y) \otimes (g(x,y;\sigma_c) - g(x,y;\sigma_s))\}, \tag{11}$$

$$SS_{off}(x,y) = \sum_{(i,j) \in surround} S_{off}(i,j), \tag{12}$$

$$R_{off}(x,y) = \mathbb{H}\{I_d(x,y) - w_{off}(x,y) \cdot SS_{off}(x,y)\}, \tag{13}$$

with

$$w_{off}(x,y) = m/CC_{off}(x,y), \tag{14}$$

where m is a (global) constant same as the above, and $CC_{off}(x,y)$ is the mean of $I_d(x,y)$ within the RF (including the center and its surround) centered at $(x,y)$.

We, then, obtain the modified detail image $I_m(x,y)$ as the sum of the processed ON and OFF path signals. We integrate the ON and OFF path outputs as

$$I_m(x,y) = R_{on}(x,y) - R_{off}(x,y). \tag{15}$$

Then, we combine the compensated detail image $I_m(x,y)$ containing the high-frequency component and the reconstructed image $I_{low}(x,y)$ containing the low-frequency component as the final output $I_o(x,y)$:

$$I_o(x,y) = I_m(x,y) + I_{low}(x,y). \tag{16}$$

## 3. Experiments

In this section, we will first discuss the influence of the parameter of superpixel size on the final reconstructed image. Then, we will specifically compare the differences between the superpixel version of bilateral filtering and normal bilateral filtering. Finally, we will validate the effectiveness of the proposed model on both synthetic and real ultrasound images, in comparison with several state-of-the-art methods.

### 3.1. Influence of the Superpixel Size

As the main parameter involved in the proposed superpixel version of bilateral filtering, the superpixel size may greatly influence the final processing results. Generally speaking, segmentation with large superpixels can only capture the coarser details or edges of size larger than the superpixel size, and the finer details would be smoothed after reconstruction. In contrast, segmentation with small superpixels can keep more fine details but degrades the de-noising ability and raises the computational cost. Figure 4 shows the reconstructed image patches, derived from Figure 1, with different superpixel sizes. We can clearly see that, with an increase of superpixel size, a mosaic pattern gradually appears in the reconstructed images; which is quite similar to pseudo-Gibbs artifacts. In contrast, smaller superpixels can capture small edges well, but more noise appears. So, considering the various organs of different sizes which are contained in the UI images, we experimentally set the superpixel size to 1% of the minimal size of the horizontal and vertical image sizes, and lower than 7 pixels, in this work.

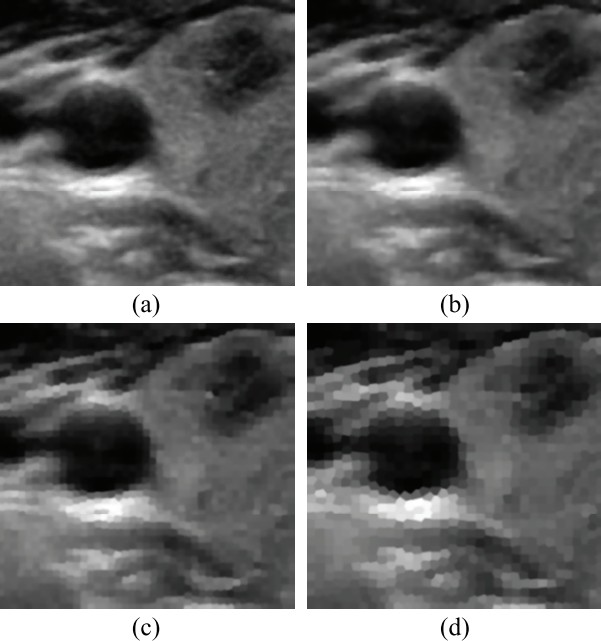

(a)          (b)

(c)          (d)

**Figure 4.** The reconstructed images with different superpixel sizes (in pixel). From (**a**–**d**), the superpixel sizes are 3, 5, 7, and 9, respectively.

### 3.2. Tests on Synthetic Images

To validate the effectiveness of the proposed strategy, we conducted our tests on synthetic images. For quantitative comparison, we first added speckle noise to the clean images by employing the synthetic speckle noise model [12] on a synthetic image, as shown in Figure 5a. As in Zhu et al. [14],

the added noise is a multiplicative Gaussian $N(0, \delta_s^2)$ noise, where $\delta_s$ controls the noise level, and we set the noise level as $\delta_s = \{0.15; 0.2; 0.25; 0.3\}$ on the test synthetic image. Figure 5b illustrates the corresponding noisy image with $\delta_s = 0.3$. In addition, we also adopted the Field II toolbox [27,28] to add speckle noise to another clean image, as shown in Figure 5c, and the corresponding image with speckle noise is shown in Figure 5d. The Field II toolbox is a program for simulating the image of ultrasound scanners. It can calculate pulsed pressure fields, and can also handle continuous wave and pulse-echo cases [27,28]. As we can see, the generated Field II image seems very close to a real ultrasound image.

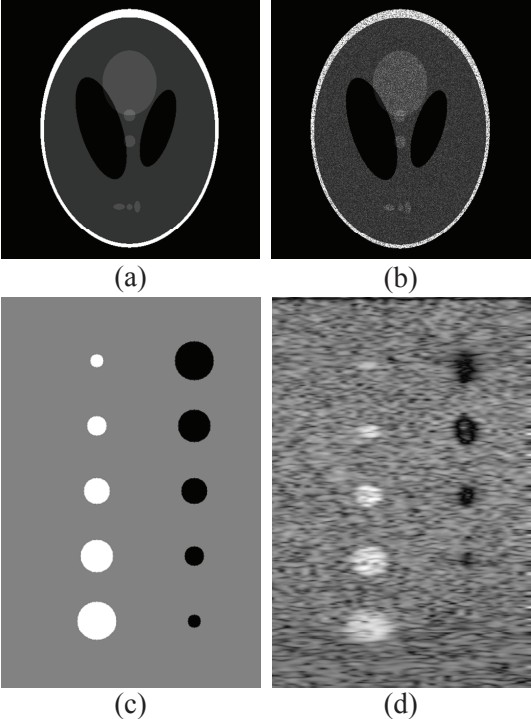

**Figure 5.** Illustration of synthetic images with speckle noise added by different methods. (**a**,**c**) are two clean images. (**b**) is a noisy image of (**a**), with multiplicative Gaussian noise (with $\delta_s = 0.3$) added, and (**d**) is a noisy image of (**c**) with speckle noise added using the Field II toolbox [27,28].

We mainly compared our method with two state-of-the-art non-local despeckling methods, the optimized Bayesian non local means filter (OBNLM) [29] and the non-local low-rank method proposed by Zhu et al. [14]. As our method employs the superpixel version of bilateral filtering, we also compared the processing results with those given by the local method of bilateral filtering [6] with fine-tuned parameters, as a baseline. We adopted two commonly used metrics, peak signal-to-noise ratio (PSNR) and structural similarity (SSIM) [30], to quantitatively compare the performance of our method against others. PSNR is an image quality measure after some modification to the image is done, and SSIM is used for measuring the structural similarity between two images. Both PSNR and SSIM are full reference metrics, which means the measurement of image quality is based on an initial distortion-free image as a reference. Generally, for the speckle reducing task, higher values of PSNR and SSIM indicate better performance.

From the PSNR and SSIM values listed in Tables 1 and 2, our method performed slightly worse than the method proposed by Zhu et al. [14] and/or OBNLM [29] on the synthetic image in Figure 5b, in which the noise was added using the multiplicative Gaussian model, but performed remarkably better on the synthetic image in Figure 5d, in which the speckle noise was added using the Field II toolbox [27,28]. We think the observation that our method performs better on the Field II-generated image is especially promising as, in comparison to the simple multiplicative Gaussian noise model,

the Field II toolbox has the ability of simulating realistic ultrasound images of human tissue [27,28]. This may enable our method to have a higher chance of better despeckling real ultrasound images in clinical practice.

**Table 1.** The peak signal-to-noise (PSNR) metric of various methods on the two synthetic images in Figure 5.

| Method | Figure 5b | | | | Figure 5d |
|---|---|---|---|---|---|
| | $\delta = 0.15$ | $\delta = 0.2$ | $\delta = 0.25$ | $\delta = 0.3$ | |
| Original | 23.35 | 22.26 | 21.41 | 21.07 | 17.73 |
| Bilateral filter | 26.32 | 24.88 | 23.73 | 23.31 | 18.77 |
| OBNLM [29] | 27.20 | 25.77 | 24.69 | 24.31 | 18.19 |
| Zhu et al. [14] | 27.10 | 26.23 | 25.56 | 25.24 | 18.64 |
| Proposed method | 26.84 | 25.59 | 24.52 | 24.13 | 19.20 |

**Table 2.** The structural similarity (SSIM) metric of various methods on the two synthetic images in Figure 5.

| Method | Figure 5b | | | | Figure 5d |
|---|---|---|---|---|---|
| | $\delta = 0.15$ | $\delta = 0.2$ | $\delta = 0.25$ | $\delta = 0.3$ | |
| Original | 0.71 | 0.69 | 0.68 | 0.68 | 0.28 |
| Bilateral filter | 0.95 | 0.95 | 0.94 | 0.93 | 0.53 |
| OBNLM [29] | 0.94 | 0.92 | 0.91 | 0.89 | 0.39 |
| Zhu et al. [14] | 0.96 | 0.96 | 0.95 | 0.94 | 0.40 |
| Proposed method | 0.95 | 0.94 | 0.93 | 0.93 | 0.61 |

We also used normal bilateral filtering followed by our compensation strategy, and no clear improvement was found over the normal bilateral filtering itself in terms of the PSNR and SSIM metrics. A possible reason is that bilateral filtering breaks the local structures, which are difficult to recover with our compensation strategy.

### 3.3. Tests on Real Ultrasound Images

Considering that the synthetic images may not fully cover all of the features of real ultrasound imaging conditions, we also conducted tests on real ultrasound images. Generally speaking, the noise levels vary for different organs. Therefore, we selected four representative ultrasound images, captured at the locations of thyroid, breast, liver, and spleen to show the generalization ability of the proposed strategy.

Figures 6 and 7 compare the processed results of the different methods. We also show zoomed-in patches for regions of interest (ROIs), which were curated by physicians for clinical diagnosis (i.e., the ROIs were picked by a physician in the department of ultrasound imaging), for each image tested here. For example, a tumor exists in the thyroid image (see the first row of Figure 6), which needs to be paid more attention to distinguish whether it is benign or not. We can see that the method of bilateral filtering can effectively reduce the speckle noise, but the processed images are obviously over-smoothed. In particular, considering that identifying calcification is very helpful for discriminating whether a tumor is benign or malignant, the over-smoothing of the images makes it difficult to identify this property of the tumor. In constrast, the images processed by the OBNLM method [29] exhibited stripe patterns, which made it unable to accurately show the real tissue situation. A possible reason for this is the inaccurate selection of similar patches in OBNLM, due to the high level of speckle noise. By contrast, our method and the method proposed by Zhu et al. [14] effectively reduced the speckle noise while retaining image details. Please note that our method preserved more details, in comparison to the method of Zhu et al. [14]. A similar phenomenon also happens for other

the organ images in Figures 6 and 7, which are consistent with the observations from the previous experiments on the synthetic images.

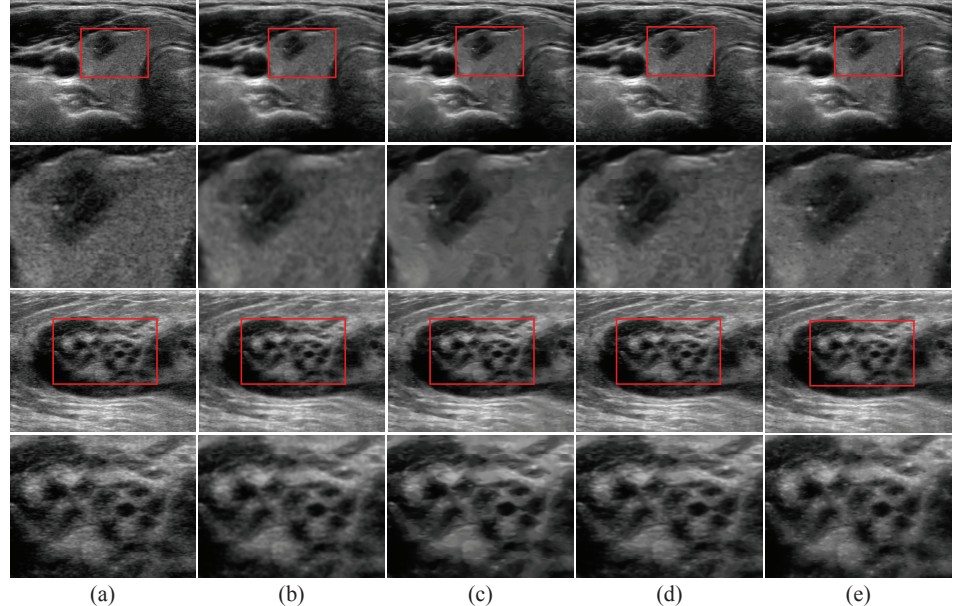

**Figure 6.** Comparison of despeckling results on the thyroid and breast ultrasound images. (**a**) The original thyroid and breast ultrasound images and the results by (**b**) bilateral filter [6], (**c**) OBNLM [29], (**d**) the method of Zhu et al. [14], and (**e**) the proposed method. The first and third rows list the full-size images, and the second and fourth rows show the zoomed-in patches containing the regions of interest (ROIs), marked by the red rectangles in the whole images.

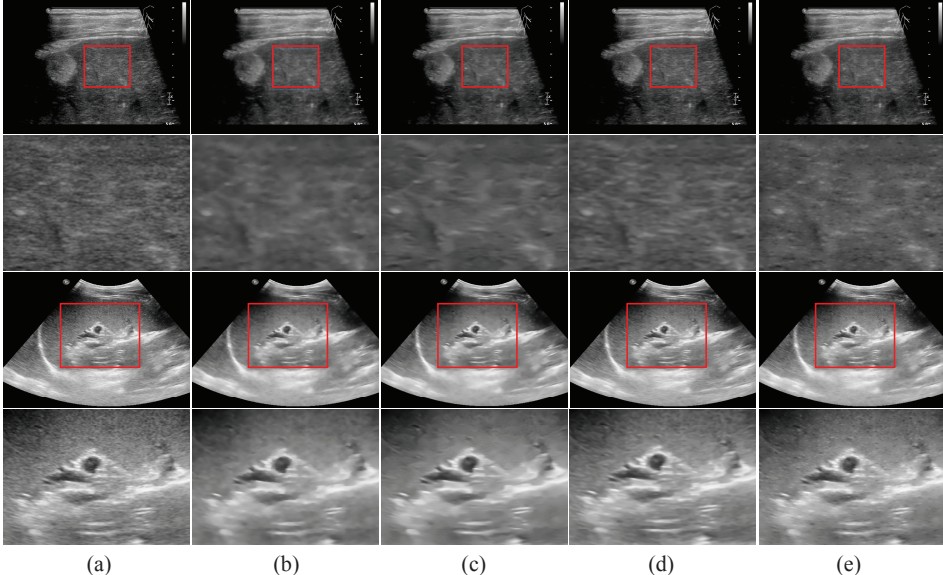

**Figure 7.** Comparison of despeckling results on the liver and spleen ultrasound images. (**a**) The original liver and spleen ultrasound images and the results by (**b**) bilateral filter [6], (**c**) OBNLM [29], (**d**) the method of Zhu et al. [14], and (**e**) the proposed method. The first and third rows list the full-size images, and the second and fourth rows show the zoomed-in patches containing the regions of interest (ROIs), marked by the red rectangles in the whole images.

## 4. Conclusions

In this paper, we proposed a speckle noise reduction (or despeckling) method for ultrasound images (UIs), based on the strategies of superpixel segmentation and spatial compensation.

In particular, the technique of superpixel segmentation was used to sparsely code the input image, which can be regarded as a superpixel version of bilateral filtering. Then, we treated the reconstructed image from the sparse representation of the superpixel segmentation as the low-frequency component, and a HVS-inspired spatial compensation was designed to fine-tune the image details that were to be recovered. Experiments on multiple synthetic and real UIs showed that the proposed method performs better than the state-of-the-art methods in terms of visual observation and objective metric evaluation.

The merits of our method can be summarized, as follows: (1) Our method employs statistic-based local processing, but does not destroy the local correlations of the details, as superpixel segmentation can effectively maintain local structures [31]. This suggests a new way for local statistic methods to provide reasonable smoothing for further processing. (2) In contrast to those traditional local methods, which may over-smooth details, a HVS-inspired image processing strategy is specifically designed in our model, in order to further compensate for and recover the image details. In particular, the center-surround opponent RF structure can effectively filter out the noise and preserve the wanted details. In addition, the ON and OFF pathways separately process the bright and dark parts, which helps to simultaneously recover the details in both the bright and dark regions of the UIs.

Please note that, differing from normal noise reduction tasks, some "speckle noise" in the ultrasound images are meaningful and are useful in clinical diagnoses. For example, liver fibrosis due to viral infection or alcohol abuse will cause the lobule to form a fibrous septum. This will lead to inhomogeneous ultrasound waves when travelling through the abnormal liver parenchyma, which will form a clinically meaningful speckle pattern in the UIs [32]. So, the compensation strategy we introduced is very important for effective speckle reduction in real applications.

Considering that superpixel segmentation is a key step in our proposed flowchart, our future works will be focused on improving its effect when applied to the specific task of UI despeckling. For example, besides the median filtering we adopted in this work before superpixel segmentation, other de-noising methods can be used, in order to achieve robust superpixel segmentation in UIs containing high-level noise. In addition, among many other superpixel segmentation methods, we can replace the SLIC method with modified superpixel segmentation methods, which may show better speckle noise tolerance abilities [31].

**Author Contributions:** Conceptualization, Y.C., M.Z., and K.-F.Y.; Funding acquisition, H.-M.Y. and Y.-J.L.; Methodology, Y.C. and M.Z.; Software, M.Z. and K.-F.Y.; Supervision, H.-M.Y. and Y.-J.L.; Validation, Y.C. and K.-F.Y.; Writing—original draft, Y.C. and M.Z.; Writing—review and editing, H.-M.Y., Y.-J.L., and K.-F.Y.

**Funding:** This research was funded by the National Natural Science Foundations of China under grants 61773094 and 61703075. The work was also supported by the Sichuan Province Science and Technology Support Project under grants 2017SZDZX0019 and 2017JY0249.

**Conflicts of Interest:** The authors declare no conflict of interest.

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
