# Peer review of "A New Ultrasound Speckle Reduction Algorithm Based on Superpixel Segmentation and Detail Compensation"

_applsci, doi:10.3390/app9081693_

Round 1

Reviewer 1 Report

Authors demonstrated a novel method, which is based on superpixel segmentation and human vision system mimicking algorithms to reduce speckle noises in ultrasound images. This manuscript is very interesting. Before publication in the journal, a few issues should be however improved.

1.    In Equation (1), σd and σr, the scale parameters, were empirically determined 4 and 0.3, respectively. I think that the parameters are crucial for the achievement of good performance in speckle reduction. It would be great to discuss the effects of the parameters on the overall performance of the algorithm. For examples, how to determine the parameters along the center frequency, bandwidth, and etc?

2.    For image compensation, authors built an nCRF model since the typical RF signals exhibited the spatial structure of the RGCs. But, the ultrasound image is constructed by envelopment of RF signals and therefore, the negative signal parts of the RF signals are diminished. Hence, I wonder whether the nCRF model is appropriate for image compensation. Please discuss these points.

3.    Figure 7, the results from the Zhu’s model is not aligned with the Figure 6. You may make a mistake. According to Table 1 and Table 2,  the Zhu’s model shows similar performance to yours. But, in Figure 7, it was not. Please check it out.

Author Response

Response to Reviewer 1 Comments

Authors demonstrated a novel method, which is based on superpixel segmentation and human vision system mimicking algorithms to reduce speckle noises in ultrasound images. This manuscript is very interesting. Before publication in the journal, a few issues should be however improved.

Point 1:  In Equation (1), σd and σr, the scale parameters, were empirically determined 4 and 0.3, respectively. I think that the parameters are crucial for the achievement of good performance in speckle reduction. It would be great to discuss the effects of the parameters on the overall performance of the algorithm. For examples, how to determine the parameters along the center frequency, bandwidth, and etc?

Response 1: Thanks for your suggestion.

It is true that the parameters (σd and σr) are crucial for image compensation and the performance in speckle reduction. In this work, we set appropriate values of σd and σr in Equation (1) to prevent the unbalance of the distance- and intensity-related weightings for image reconstruction instead of a simple normalization. This is mainly due to a technical consideration that the setting of σd and σr should depend on the superpixel size in this study. For example, according to the analysis and discussion in Section 3.1, we set the superpixel size as about 7 pixels in this work. Therefore, for the image reconstruction with four neighboring superpixels, the maximal distance between the current point and the neighboring superpixel centers (i.e., the term sqrt((i-k)2+(j-l)2) in Equation (1)) is about 10 pixels (along the diagonal line). Therefore, we set σd=4 to control the decreasing rate of the distance-related weighting. On the other hand, the maximum intensity difference between two pixels (i.e., the term ||I(i,j)-I(k,l)|| in Equation (1)) is 1.0. Therefore, we set σr=0.3 to control the decreasing rate of the intensity-related weighting.

In summary, the setting of σd=4 and σr =0.3 used in this work can well balance the distance- and intensity-related weightings (i.e., with similar decreasing rates) when reconstructing the image with the superpixel size as about 7 pixels. Note that as mentioned above, the setting of σd and σr should be dependent on the superpixel size. Large σd and σr values are required for the image segmentation with large superpixel sizes, and vice versa.

The above explanation is briefly clarified in the second and third paragraph of Section 2.2.

Point 2: For image compensation, authors built an nCRF model since the typical RF signals exhibited the spatial structure of the RGCs. But, the ultrasound image is constructed by envelopment of RF signals and therefore, the negative signal parts of the RF signals are diminished. Hence, I wonder whether the nCRF model is appropriate for image compensation. Please discuss these points.

Response 2: Thanks for your insightful comment. We believe that our nCRF model is appropriate for image compensation.

    In this study, the nCRF model is used to adaptively separate the useful details from the unwanted speckle noises contained in the high-frequency component. According to Equation (3), the high-frequency component of an ultrasound image contains both positive and negative signal parts. Accordingly, we first separate the high-frequency component into the bright and dark parts (i.e., the positive and negative signal parts, see Equation (4) and Equation (5)), and then employ different nCRF models (i.e., ON-type and OFF-type) to reduce the speckles while preserving the useful details.

   Note that we only diminish the negative signal parts when computing the surround inhibition in ON- or OFF-pathway independently. For example, in the ON-pathway, we use a rectification operator (which diminishes the negative signal parts) to guarantee that neuronal responses should not be negative (see Equation (7) and Equation (9)). However, according to the nCRF model, obtaining negative signals after surround inhibition means that these pixels receive strong inhibition, and hence they should be regarded as speckles and be removed. Therefore, setting the negative responses to zeros can not only achieve despeckling, but also avoids the further processing of those potential interference of negative signals.

The above explanation is briefly clarified in the sixth paragraph of Section 2.3 (after Equation (9)).

Point 3:  Figure 7, the results from the Zhu’s model is not aligned with the Figure 6. You may make a mistake. According to Table 1 and Table 2, the Zhu’s model shows similar performance to yours. But, in Figure 7, it was not. Please check it out.

Response 3: Thanks for your comment. We are sorry for this mistake on the results of Zhu's model in Figure 7. In the revised manuscript, we have updated all the results in the experimental section.

    In addition, the performance listed in Table 1 and Table 2 was tested on two synthetic images shown in Fig. 5, where Fig. 5(b) is a noisy image with multiplicative Gaussian noises and Fig. 5(d) is a noisy image with speckle noises added using the Field II toolbox. From Table 1 and Table 2, we can see that Zhu’s model shows better performance than our model on the Gaussian noisy image of Fig. 5(b); while our model performs better than Zhu’s model on the synthetic speckle image of Fig. 5(d) (see the last column in Table 1 and Table 2). This indicates that our model has a higher chance to better despeckle the real ultrasound images. These results are consistent with the results in Fig. 6 and Fig. 7 (of the revised version).

Reviewer 2 Report

This paper proposes an algorithm for ultrasound speckle reduction based on a super-pixel version of bilateral filtering in order to preserve the local structures during denoising.

The topic is interesting and is on point for this journal as similar articles have recently been published here [1]. The area is well-researched but a more up-to-date survey paper might be appropriate, e.g. [2].

The method outlined is nice but there are a few minor issues with its presentation in the paper that can be easily addressed:

The weighting system for reconstruction may cause issues as described as it combines distances and intensities - are these normalised to prevent this? If so, this should be clarified.

The pre-processing step (median filtering) is left out of Figure 1 and not discussed further. However, this could have a significant effect on the outcome. The median filter used should be briefly outlined (e.g. the kernel size would be important).

For the weightings used, the authors state “We experimentally set won = 1 and woff = 0.6 in this work”. What does this mean? Trail and error? Testing on how many images with what diversity?

PSNR and SSIM should be introduced, at least in so far as high PSNR & SSIM good, low bad.

There is a typo on Table 1: “PNSR”. Also, “The proposed” should read “Proposed method”.

There is a typo throughout the paper “Filed II Toolbox”. Also, this should be introduced.

The English is generally good and the paper very readable but a glance over by a native English speaker would help pick up on minor language issues.

Really should have these two references added to the end:
[1]    M. Y. Jabarulla and H.-N. Lee, ‘Speckle Reduction on Ultrasound Liver Images Based on a Sparse Representation over a Learned Dictionary’, Appl. Sci., vol. 8, no. 6, p. 903, Jun. 2018.
[2]    T. Joel and R. Sivakumar, ‘An extensive review on Despeckling of medical ultrasound images using various transformation techniques’, Appl. Acoust., vol. 138, pp. 18–27, 2018.

Author Response

Response to Reviewer 2 Comments

This paper proposes an algorithm for ultrasound speckle reduction based on a super-pixel version of bilateral filtering in order to preserve the local structures during denoising.

The topic is interesting and is on point for this journal as similar articles have recently been published here [1]. The area is well-researched but a more up-to-date survey paper might be appropriate, e.g. [2].

The method outlined is nice but there are a few minor issues with its presentation in the paper that can be easily addressed:

Point 1: The weighting system for reconstruction may cause issues as described as it combines distances and intensities - are these normalised to prevent this? If so, this should be clarified.

Response 1: In this work, we set appropriate values of σd and σr in Equation (1) to prevent the unbalance of the distance- and intensity-related weightings for image reconstruction instead of a simple normalization. This is mainly due to a technical consideration that the setting of σd and σr should depend on the superpixel size in this study. For example, according to the analysis and discussion in Section 3.1, we set the superpixel size as about 7 pixels in this work. Therefore, for the image reconstruction with four neighboring superpixels, the maximal distance between the current point and the neighboring superpixel centers (i.e., the term sqrt((i-k)2+(j-l)2) in Equation (1)) is about 10 pixels (along the diagonal line). Therefore, we set σd=4 to control the decreasing rate of the distance-related weighting. On the other hand, the maximum intensity difference between two pixels (i.e., the term |I(i,j)-I(k,l)| in Equation (1)) is 1.0. Therefore, we set σr=0.3 to control the decreasing rate of the intensity-related weighting.

In summary, the setting of σd=4 and σr =0.3 used in this work can well balance the distance- and intensity-related weightings (i.e., with similar decreasing rates) when reconstructing the image with the superpixel size as about 7 pixels. Note that as mentioned above, the setting of σd and σr should be dependent on the superpixel size. Large σd and σr values are required for the image segmentation with large superpixel sizes, and vice versa.

The above explanation is briefly clarified in the second and third paragraph of Section 2.2.

Point 2: The pre-processing step (median filtering) is left out of Figure 1 and not discussed further. However, this could have a significant effect on the outcome. The median filter used should be briefly outlined (e.g. the kernel size would be important).

Response 2: In the pre-processing step, the kernel size of the median filter is 3×3. The purpose of this pre-processing is to reduce the influence of high-frequency noises (not speckles) and improve the robustness of SLIC segmentation. Therefore, we set the median filter to a small size; namely, 3×3.

We have briefly outlined of median filtering in the first paragraph of Section 2.1 and mentioned this step in Figure 1.

Point 3: For the weightings used, the authors state “We experimentally set won = 1 and woff = 0.6 in this work”. What does this mean? Trail and error? Testing on how many images with what diversity?

Response 3: Yes, here “experimentally” means “empirically” or “by trail-and-error”.

Equation (15) is used to combine the high-frequency components processed along the ON- and OFF-pathway in a weighted way (with won and woff as the weightings). We have re-checked the computing process and found that the proposed method obtains similar results when we set won = woff  = 1.0, compared to the results in the original experiments with won =1 and woff =0.6. Therefore, we have safely revised Equation (15) as "Im(x, y) = Ron(x, y) - Roff (x, y)" in this revised version to reduce the number of free parameters by removing these two weightings. We have also updated all the results in the experimental section.

Point 4: PSNR and SSIM should be introduced, at least in so far as high PSNR & SSIM good, low bad.

Response 4: PSNR is an image quality measure after some modification to the image is done, and SSIM is used for measuring the structural similarity between two images. Both PSNR and SSIM are full reference metrics, which means the measurement of image quality is based on an initial distortion-free image as a reference. Generally, for the speckle reducing task, higher values of PSNR and SSIM indicate better performance.

We have added a brief introduction to PSNR and SSIM in the second paragraph of Section 3.2.

Point 5: There is a typo on Table 1: “PNSR”. Also, “The proposed” should read “Proposed method”.

Response 5: We are sorry for this typo. We have revised "PNSR" to "PSNR", and "The proposed" to "Proposed method" in Table 1.

Point 6: There is a typo throughout the paper “Filed II Toolbox”. Also, this should be introduced.

Response 6: We are sorry for this typo. We have revised "Filed II Toolbox" to "Field II Toolbox" and checked other same typos throughout the paper.

Field II Toolbox is a program to simulate the imaging process of ultrasound scanners. It can calculate the pulsed pressure fields, and also handle the continuous wave and pulse-echo cases. We have added a brief introduction to the Field II Toolbox in the first paragraph of Section 3.2.

Point 7: The English is generally good and the paper very readable but a glance over by a native English speaker would help pick up on minor language issues.

Response 7: We have asked the MDPI English editing service (https://www.mdpi.com/ authors/english) to help refine and polish the English usage of the revised manuscript.

Point 8: Really should have these two references added to the end:
[1] M. Y. Jabarulla and H.-N. Lee, ‘Speckle Reduction on Ultrasound Liver Images Based on a Sparse Representation over a Learned Dictionary’, Appl. Sci., vol. 8, no. 6, p. 903, Jun. 2018.
[2] T. Joel and R. Sivakumar, ‘An extensive review on Despeckling of medical ultrasound images using various transformation techniques’, Appl. Acoust., vol. 138, pp. 18–27, 2018.

Response 8: These two references have been cited in the appropriate positions of the revised paper.
